# Multilayer scanning enhances sensitivity of artificial intelligence-aided Mycobacterium tuberculosis detection

**Yan Xiong**
Peking University First Hospital, China.
`yanxiong1109@163.com`

**Ao Hou**
Shenzhen Semptian Co., Ltd., China.
`ao_sure@foxmail.com`

**Ting Li**
Peking University First Hospital, China.
`lixiaoting12@hotmail.com`

**Longsen Chen**
Shenzhen Semptian Co., Ltd., China.
`chenlongsen@semptian.com`

**Lifang Chen**
Shenzhen Semptian Co., Ltd., China.
`chenlifang@semptian.com`

**Lili Lai**
Shenzhen Semptian Co., Ltd., China.
`lailili@semptian.com`

## Abstract

In the study [1] of automatic detection of Mycobacterium Tuberculosis (TB) using artificial intelligence, 201 samples (108 positive cases and 93 negative cases) were collected as a test set and used to examine TB-AI and TB-AI achieved $97.94\%$ sensitivity and $83.65\%$ specificity. However, with single-layer scanning, some Mycobacterium TBs are blurred due to the defocus. As a result, slides with blurred TB pixels may not be detected as positive In this paper a new test of TB-AI with three-layer scanning was conducted on 189 positive cases reported by medical doctors with microscope. Comparing to the ordinary single-layer scanned slides, additional 6 out of 189 cases $(3.2\%)$ were detected.

## 1 Introduction

Pathology is one of the most important means for diagnosing TB in clinical practice. To confirm TB as the diagnosis, finding specially stained TB bacilli under a microscope is critical. Upon the acid-fast staining[2], The waxy lipid in the cell wall of the bacilli appears purple red after acid-fast staining, showing high contrast to the blue background[3]. Detecting bacilli with such morphology and color upon staining is specific to the diagnosis of TB. However, Because of the very small size (less than 1 $\mu$m in diameter) of the bacilli, to look for and identify them under the microscope requires use of high-power fields, which provide a rather limited visual area over a whole tissue section. Besides, the number of bacilli is usually small. Therefore, it is a time-consuming and strenuous work even for experienced pathologists, and this strenuosity often leads to low detection rate and false diagnoses. In order to improve the efficiency and sensitivity of the detection of TB bacilli, several new techniques have been developed, including PCR and RNA scope, but so far none of them have proved to be reliable and accepted widely[4].

The computer-based artificial intelligence (AI) [5] was employed to perform the task of detection and identification of digital scanning image of Mycobacterium TB, which can reduce the subjectivity of medical diagnosis and reduce heavy burden of doctors. Thus, it has a bright prospect in the practical clinical application. Some of research and achievements were illustrated in the literature on the

1st Conference on Medical Imaging with Deep Learning (MIDL 2018), Amsterdam, The Netherlands.

automatic detection of mycobacterium tuberculosis using artificial intelligence[1]. As for the CAD of acid-fast staining Mycobacterium TB, the sensitivity is $97.94\%$, and the specificity is $83.65\%$[1].

Based on this research, we conducted a lot of experiments and explorations, hoping to further improve the overall performance of CAD of Mycobacterium TB. We collected more data on slides from different hospitals, and performed the task of data annotation in a more rigorous manner under the supervision of pathologists. In addition, we cleaned those unqualified samples, and added more samples for training and test, and improved samples, thereby enhancing the coverage and diversity of samples. As for the model, we employed the model of convolutional neural network with characteristics of optimized functions and complex structures, and we performed the tasks of algorithm optimization and model upgrade. Notwithstanding some improvements in our experimental results thanks to these efforts, so far, remarkable progress has not been registered. Therefore, the work mentioned above is not illustrated in this paper.

As for the research in the early phase, we directed large amounts of our energies to the above-mentioned processes. Thus, no research on the digital scanning of tissue slides was conducted. During the multiple rounds of model finetune and test, some of the slides which had been diagnosed as positive according to the results displayed the microscope were diagnosed as consistently negative by means of computer-based artificial intelligence. Through a series of comparison and verification by pathologists, results were found that some Mycobacterium TB has no clear image on digital slides, or has completely disappeared. This phenomenon has aroused our attention. We conducted an analysis of the new dataset of digital slides and found that most of the digital slides have some Mycobacterium TB characterized with unclear image. Undoubtedly, this problem is helpful for enhancing the sensitivity of CAD. Therefore, the following studies were conducted for the missed detection of Mycobacterium TB caused by defocus-oriented unclear image or disappearance in the background upon the slide scanning.

## 2   Related technologies

All slides were scanned using a KF-PRO-005 Digital Section Scanner (Ningbo Jiangfeng Bio-information Technology Co., Ltd., Ningbo, China). The device, with a total of 5 slides at a time, boasts its high-precision, top-quality digital Whole slide image (WSI). It employs a professional 3CCD linear camera, and, thanks to three primary colors, it is equipped with an independent color processing channel, thus securing natural, accurate color reproduction and clear image. It is characterized with the one-button operation, which is simple and convenient.

The digital slide scanner, by default, adopts the scanning mode of single-layer automatic adaptive focusing. It, as shown in Figure 1(b), can automatically select a limited number of target points in the slide, as the reference for the focusing task. Then, it automatically optimizes and adjusts the focusing distance of the scanning process. It, compared to the mode of employing a single, fixed focusing distance, has indeed enhanced the imaging quality. However, in order not to seriously affect the rate and of scanning, the focusing distance of the scanning process cannot be adjusted on a frequent basis. Therefore, this mode of operation suffers from certain drawbacks[6].

As is shown in Figure 1(a), it is one of the simplest methods of multilayer scanning. First, determine a fixed focusing plane as the reference; obtain multiple focusing planes to be scanned by deviating the focusing plane up and down by a fixed distance[the deviation in Figure 1(a) is 0.8 $\mu$m][a three-layer focusing plane is shown in Figure 1(a)]; perform multiple tasks of focusing-changed scanning, in order to obtain multiple slide images which were scanned.

Figure 1(c) shows the operating principle of multilayer scanning aided by the KF-PRO-005 Digital Section Scanner. This paper employs the three-layer scanning - based on the principle of single-layer automatic adaptive focusing, the focusing plane of the lens is deviated up and down by a fixed distance (the deviation in Figure 1(c) is 0.8 $\mu$m); three layers of image scanning are formed, each corresponding to a piece of digital slide image.

Prior to this study, we tested and evaluated the multilayer scanning effect of the KF-PRO-005 Digital Section Scanner. Based on this, we determined to use the three-layer scanning mode with the space in between of 0.8 $\mu$m.

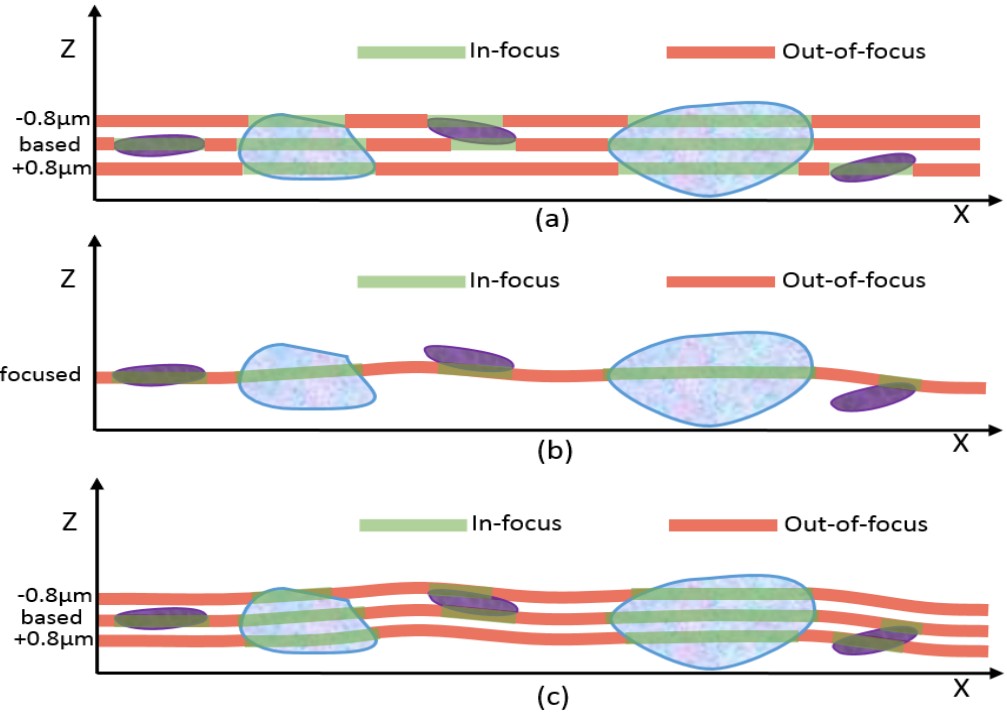

Figure 1: shows the cross-sectional view of three identical slides. The thick line indicates the scanning layer, the green line indicates the focus area, and the red line indicates the non-focus area. (a) shows the principle of fixed-focus multilayer scanning, with the space of focus planes in between of 0.8 $\mu$m; (b) shows the principle of focus-adaptive single-layer scanning which was improved; (c) shows the principle of focus-adaptive multilayer (three-layer) scanning, with the space in between of 0.8 $\mu$m.

## 3   Method

1. As for this study, a total of 189 cases with positive acid-fast staining results were collected from Department of Pathology, Peking University First Hospital during January 2013 to December 2017.

2. All specimens were processed in strict accordance with the standard procedures of tissue fixation, embedding, slide preparation, and acid-fast staining.

3. All the slides to be tested were reviewed by two specific pathologists by means of a 40× magnification microscope. Acid-fast staining Mycobacterium TB was identified on all the slides according to the results displayed by the microscope. The re-detection results of all the slides to be tested were consistent with the original records produced by the hospital - positive.

4. All slides were scanned using a KF-PRO-005 Digital Section Scanner (Ningbo Jiangfeng Bio-information Technology Co., Ltd., Ningbo, China). Configuration: the three-layer scanning mode with 0.8 $\mu$m interlayer spacing. Upon on the scanning, a total of 567 (3×189) digital slides were obtained.

5. In the experiment, the mid layer of each set of three layers was used as the data for single-layer scanning.

6. As for all the digital slides obtained from 189 test samples, we by means of the artificial intelligence TB-AI[1] realized in the previous studies, conducted a diagnosis test on acid-fast staining Mycobacterium TB.

7. Then, we collected the experimental results, and performed such tasks as statistical analysis and comparison. We built on work to make a study on the effect brought about by the three-layer scanning technique to acid-fast staining of Mycobacterium TB aided by artificial intelligence.

Table 1: TB-AI Test results of acid-fast staining Mycobacterium TB under different scanning modes. A total of 207 positive cases were confirmed (TB can be identified artificially according to the results displayed by the microscope)

| Scanning method | Description | TB Detected Rate | TB Undetected Rate |
|---|---|---|---|
| Single-layer scanning | Automatic focusing | **177 (93.7%)** | 12 (6.3%) |
| Three-layer scanning | Layer_1: offset 0.8 $\mu$m upwards | 173 (91.5%) | 16 (8.5%) |
| | Layer_0: Automatic focusing | 177 (93.7%) | 12 (6.3%) |
| | Layer_-1: offset 0.8 $\mu$m downwards | 176 (93.1%) | 13 (6.9%) |

Table 2: TB-AI statistical results of the three-layer scanning conditions for the test of acid-fast staining Mycobacterium TB

| Condition | TB Detected Rate | TB Undetected Rate |
|---|---|---|
| All the three layers | 168 (88.9%) | 21 (11.1%) |
| At least one of the three layers | **183 (96.8%)** | 6 (3.2%) |

## 4 Results

A total of 189 cases of acid-fast staining Mycobacterium TB were used for the test. Each of the slides was re-diagnosed by two pathologists by means of a 40× magnification microscope. TB was identified on all the slides according to the results displayed by the microscope. The re-detection results of all the slides to be tested were consistent with the original records produced by the hospital - positive.

As for the digital slides with the principle of single-layer automatic focusing-adaptive scanning, 177 cases of TB was detected by TB-AI, accounting for 93.7% of all 189 test cases, and no TB was identified in the other 12 cases (6.3%). (See Table 1).

As for digital slides with the principle of automatic focusing-adaptive three-layer scanning (ie. the mid scanning layer), 177 cases of acid-fast staining Mycobacterium TB was detected by TB-AI, accounting for 93.7% of all 189 test cases, and no TB was identified in the other 12 cases (6.3%). As for the digital slides characterized with the focusing layer deviating upwards by 0.8 $\mu$m from the scanning layer (ie. the upper scanning layer), 173 cases of TB was detected by TB-AI, accounting for 91.5% of all the test cases, and no TB was identified in the other 16 cases (8.5%). As for the digital slides characterized with the focusing layer deviating downwards by 0.8 $\mu$m from the scanning layer (ie, the upper scanning layer), 176 cases of TB was detected by TB-AI, accounting for 93.1% of all the test cases, and no TB was identified in the other 13 cases (6.9%). (See Table 1).

As for the three-layer scanning, we made an analysis of the three layers of digital slides of the same case. According to the results, 168 cases of TB were detected in all the three layers by TB-AI, accounting for 88.9%. 183 cases of TB were detected in at least one of the three layers, accounting for 96.8%. (See Table 2).

## 5 Discussion

1. The scanning device has a principle which is similar to that of ordinary optical microscopes -the microscope's lens has a small depth of field of view. The tissue slide has a certain degree of thickness, and the contents of the sample tissue are not even and regular. In particular, Mycobacterium TB (about 4 $\mu$m in length, with its diameter less than 1 $\mu$m), compared with the tissue cells, is relatively small, and, on the same focusing plane, it is easy to produce unclear image due to the defocus. For those slides with very little amount of Mycobacterium TB, it's more than likely that the image is unclear due to the defocus of Mycobacterium TB. As a result, the computer makes a negative detection on such digital slides, thus resulting in the missed detection of positive results.

2. We made a comparison of the digital images obtained by the three-layer scanning, and found that the automatic focusing-adaptive technology of the scanning device still has certain setbacks. Specifically, a three-layer scanning screenshot of the three fields of view was selected (See Figure 2). Among the three different fields of view, Figure 2(a), 2(b) and 2(c) represent the same field of view: relatively speaking, 2(a) is the clearest, followed by 2(b) and 2(c). Figure 2(d), 2(e) and 2(f) represent the same field of view: relatively speaking, 2(e) is the clearest, followed by 2(d) and 2(f). Figure 2(g), 2(h) and 2(i) represent the same field of view: relatively speaking, 2(i) is the clearest, followed by 2(g) and 2(h).

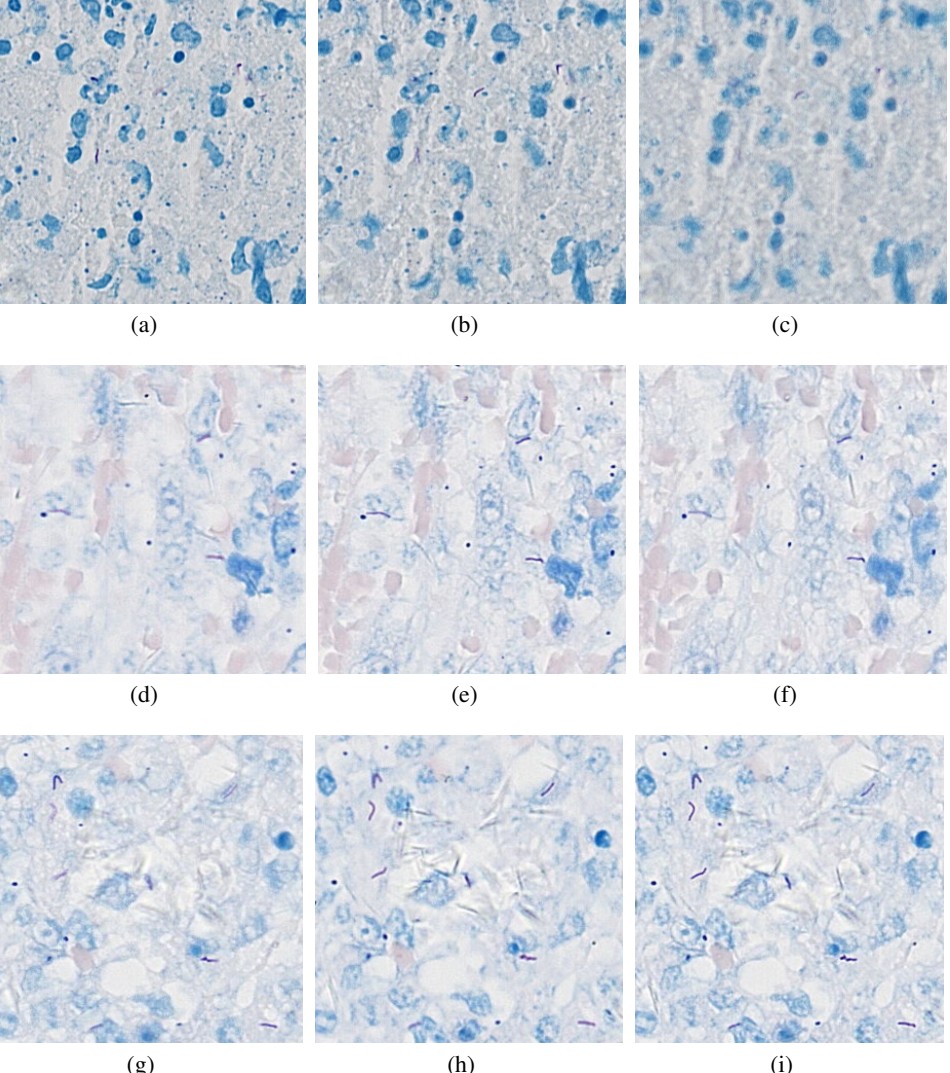

Figure 2: shows the three-layer scanning image (the focusing plane with the space in between of 0.8 $\mu$m) in three fields of view. (b) (e) (h) are the images, by default, on the reference focusing plane for the three fields of view respectively. (a) (d) (g) have their focusing planes upwards by 0.8 $\mu$m, while (c) (f) (i) have their focusing planes downwards by 0.8 $\mu$m. Obviously, among the images of different focusing planes in the same field of view, the resolution of targeted Mycobacterium TB differs remarkably.

3. Figure 2 shows the actual automatic focusing scanning layers of the three fields of view 2(b), 2(e) and 2(h): 2(b) and 2(h) are not actually the clearest in the corresponding field of view. On the different scanning layers in the same field of view, TB was shown in both screenshots 2(a) and 2(b), distributed at different locations which can be seen clearly. Therefore, the multilayer scanning can

really make up for the incomprehensive number of targeted Mycobacterium imaging caused by the single-layer scanning.

4. Since the microscopic imaging instrument suffers from a relatively small depth of field of view, the tiny Mycobacterium TB is more likely to produce an unclear image [Figure 2(c)] or even disappear [Figure 2(g)] due to the minor defocus (0.8 $\mu$m in this experiment). Upon the multilayer scanning, the distance between the focusing planes of the microscope's lens cannot be too large. According to the practical reality of our experiment, it is recommendable to configure the three layers with the space in between of 0.8 $\mu$m. However, we can also shorten the space in between, for the sake of the multilayer scanning, for example: 0.5 $\mu$m is set for the sake of the five-layer scanning.

5. The three-layer scanning can make up for some defects caused by the single-layer scanning. However, the three-layer scanning, likewise, also suffers from the problem that no valid scanning data can be obtained in case of focusing failure of the device.

6. Based on 189 cases which were diagnosed as positive according to the results displayed by a microscope, the sensitivity is $93.65\%$ under the mode of single-layer scanning. Under the mode of three-layer scanning, the highest sensitivity is $96.83\%$ (Table 2: TB detected in at least one layer of the three-layer scanning image of the same slide). It can be concluded that the three-layer scanning, compared with the ordinary single-layer scanning, can indeed facilitate TB-AI in reducing the rate of missed detection of positive cases and enhancing the sensitivity of detection. As for this experiment, the rate of missed detection of positive cases was lowered down by $3.2\%$.

7. In the previous studies of Xiong Y. et al., 201 cases (108 positive cases and 93 negative cases) were employed for the test, achieved $97.94\%$ sensitivity[1]. However, in this experiment, we made a test on the 189 new positive cases. The highest degree of sensitivity is only $96.83\%$ when the method of three-layer scanning was employed to optimize the experimental results. Indirectly it shows that TB-AI does not really attain the desirable higher degree of sensitivity, which needs continuous optimization and promotion in the future.

8. Upon the multilayer scanning, multiple files on the digital slides are generated, which have increased the amount of data. As a result, more resources of data storage are occupied. The detection also needs more calculation time and resources. More time, as well as resources of storage and calculation, is needed, accompanying the lowered rate of missed detection.

## 6 Conclusion

In this study, we using the method of three-layer scanning to generate three $40\times$ magnification digital slide images of the same case. Then, we made a comparison on the single-layer scanning results generated by default, and it showed that due to the three-layer scanning, the rate of missed detection has reduced by $3.2\%$. The tiny Mycobacterium TB is more likely to produce an unclear image or even disappear due to the defocus, thus resulting in a certain degree of missed detection, The multilayer scanning makes up for the setback caused by the single-layer scanning to some extent. As a result, it has enhanced the sensitivity of artificial intelligence in the auxiliary diagnosis of acid-fast staining Mycobacterium TB.

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
