# OpenReview forum: "Multilayer Scanning Enhances Sensitivity Of Artificial Intelligence-aided Mycobacterium Tuberculosis Detection"
_MIDL.amsterdam/2018/Conference — Submitted to MIDL 2018_

### Review · AnonReviewer1 · 2018-05-06
**An extension of a previous work to combine results from several scanning layers by a simple rule**

**Rating:** 1
**Confidence:** 3

**Review:**

This paper applies a convolutional neural network classifier developed in previous work [1] (TB-AI) for detecting tuberculosis in microscopy images to a new dataset. Unlike in [1], images are captured on 3 focal layers instead of using a single focus, and thus 3 classification results can be obtained for each sample. Experiments show that false negatives are reduced by combining the 3 classification results for each sample on a dataset that only contains positive samples.

This paper presents no methodological contribution in terms of the machine learning algorithm, since a classifier developed in [1] was directly used without any extra modification or fine-tuning. The novelty of this paper therefore lies entirely on using a new dataset that contains 3 focal layers for each sample instead of just 1. The classification result of the 3 layers is then combined with a maximum operation. In addition to this being a very small methodological contribution, the experimental validation is insufficient given that only positive samples are used. With the current results, it is impossible to know whether there is any increase of false positive detections with the new approach.

**Special Issue:**

No

---

### Review · AnonReviewer3 · 2018-05-07
**Application of a previously proposed method to Mycobacterium TB detection from multi-layer scanning images**

**Rating:** 1
**Confidence:** 2

**Review:**

Summary:
This paper presents a method to enhance the sensitivity of automatic Mycobacterium TB detection on digital scanning images based on AI. They utilize multi-layer scanning images to improve detection errors caused by the blurred images, which are occasionally included in the slide images. They achieved the detection rate of 96.8% on 189 positive cases.

Major comments:
This paper deals with a relevant research topic. However, the paper has many problems to be improved.
-The major concern is lack of technical novelty. In terms of the AI algorithm for Mycobacterium TB detection, the previously proposed method is directly used.
-Another concern is whether the AI-based method is better for Mycobacterium TB detection compared to other approaches. Although this study focuses on the input data to the TB-AI for improvement of its sensitivity, the authors should describe the reason why they adopted an AI-based method for Mycobacterium TB detection. I think detection methods considering the color information, such as [ref1], work better because the bacilli seem to have a color different from background in the images after the acid-fast staining.
[ref1] Costa Filho, Cicero Ferreira Fernandes, et al. "Automatic identification of tuberculosis mycobacterium." Research on Biomedical Engineering 31.1 (2015): 33-43.
-Although the authors used only positive samples to validate the performance of TB-AI, how well does the TB-AI work on the images of negative cases? To accurately evaluate the sensitivity for detection of Mycobacterium TB, shouldn’t the performance be validated on samples including both positive and negative cases as in the previous study [1]?
-The description regarding the training of TB-AI is not given in this manuscript. How was the TB-AI trained? How many samples were used for training?
-Overall, I found this paper difficult to read. In particular, I found “Method” and “Discussion” sections difficult to follow. The itemization in these sections has little meaning. For example, it may be helpful to itemize a summary of the workflow of method to help orient the reader.
-I would recommend that the authors describe what is included and what needs to be detected in the scanning images while showing the actual images in an earlier section (e.g., in the introduction section) to clarify the problem to be solved.
-Please describe the size and the resolution of images used for training and testing.

Minor comments:
-Please follow the elementary rules for academic writing. For example, it is better to avoid citing any references and using abbreviations in the abstract section. Even if abbreviations are introduced in the abstract, they should be spelled out and redefined at their first mention in the main text.
-Please describe what is focused on the focusing plane in Figure 1.
-Please review typos and grammar expression because there are a number of typographical and grammatical errors throughout the manuscript.

**Special Issue:**

No

---

### Review · AnonReviewer2 · 2018-05-08
**Very unclear manuscript which makes it difficult to judge the contributions made by the authors**

**Rating:** 1
**Confidence:** 3

**Review:**

The manuscript is written in a very unclear way and with highly unusual formatting (e.g. numbered paragraphs), which makes it very difficult to judge the work by the authors. In particular, it is unclear what is the additional contribution w.r.t. the work published in ref [1], and from what little I can gather there is no methodological novelty.

**Special Issue:**

No

---

### Decision · Program_Chairs · 2018-05-15
**Paper30 Acceptance Decision**

Reject